# TOKEN-BASED AUDIO INPAINTING VIA DISCRETE DIFFUSION

**Tali Dror**[*,1]  **Iftach Shoham**[*,2]  **Moshe Buchris**[1]  **Oren Gal**[3]
**Haim Permuter**[1]  **Gilad Katz**[2]  **Eliya Nachmani**[1]

[1]School of Electrical and Computer Engineering, Ben-Gurion University of the Negev
[2]Faculty of Computer and Information Science, Data Science Research Center,
Ben-Gurion University of the Negev   [3]University of Haifa
`{talidr, iftachs}@post.bgu.ac.il, eliyanac@bgu.ac.il`

## ABSTRACT

Audio inpainting seeks to restore missing segments in degraded recordings. Previous diffusion-based methods exhibit impaired performance when the missing region is large. We introduce the first approach that applies discrete diffusion over tokenized music representations from a pre-trained audio tokenizer, enabling stable and semantically coherent restoration of long gaps. Our method further incorporates two training approaches: a derivative-based regularization loss that enforces smooth temporal dynamics, and a span-based absorbing transition that provides structured corruption during diffusion. Experiments on the MusicNet and MAESTRO datasets with gaps up to 750 ms show that our approach consistently outperforms strong baselines across range of gap lengths, for gaps of 150 ms and above. This work advances musical audio restoration and introduces new directions for discrete diffusion model training. Visit our *project page* for examples and code.

## 1 INTRODUCTION

Audio inpainting refers to the task of reconstructing missing or corrupted segments of an audio signal (Le Roux et al., 2008; 2011). It is a fundamental inverse problem in audio processing, with applications ranging from restoring damaged recordings and removing artifacts, to filling in gaps caused by data loss in transmission or editing operations (Moliner & Välimäki, 2023). Traditional approaches to this problem often relied on signal modeling techniques such as autoregressive models (Adler et al., 2011), sparse representations (Marafioti et al., 2020), or linear predictive coding (Adler et al., 2011). While effective under certain assumptions (e.g., local stationarity), these methods typically perform well only on short gaps and may struggle with long-range dependencies or semantic coherence (Moliner & Välimäki, 2023; Zhou et al., 2019).

In recent years, deep generative models have significantly advanced the state of audio inpainting. Models such as Variational Autoencoders (VAEs) (Marafioti et al., 2019) and diffusion probabilistic models (Kong et al., 2020; Liu et al., 2023; Ho et al., 2020) have demonstrated the ability to learn expressive priors directly from large-scale audio datasets. Notably, diffusion models have emerged as particularly powerful tools for solving ill-posed inverse problems due to their iterative denoising process and strong generative capacity. Approaches such as DiffWave (Kong et al., 2020) apply diffusion directly on waveform samples, while others such as MAID (Liu et al., 2023) and CQT-Diff+ (Moliner & Välimäki, 2023) operate in the continuous time-frequency domain using spectrograms or Constant-Q Transform (CQT) representations (Moliner & Välimäki, 2023; Zhou et al., 2019).

While deep generative approaches have achieved strong results, they each face challenges on long gaps. Waveform-level diffusion models (e.g., DiffWave) operate at the raw sample rate, making it difficult to capture long-range structure without very large receptive fields. Spectrogram and CQT-

---

[*]Equal contribution

based methods (e.g., MAID, CQT-Diff+) use efficient time-frequency representations but depend on phase reconstruction and often lose coherence across extended regions. Similarly, VAE-based models preserve local continuity but struggle to maintain semantic consistency as the gap length increases. These limitations highlight the difficulty of reconciling fine temporal detail with high-level structure in continuous audio representations.

In this work, we propose a novel approach: *Audio Inpainting via Discrete Diffusion* (AIDD[1]). Unlike prior methods that operate in the continuous domain (Kong et al., 2020; Liu et al., 2023; Moliner & Välimäki, 2023; Zhou et al., 2019), our method applies the diffusion process to a discrete latent space. Specifically, we employ the *WavTokenizer* (Ji et al., 2024) to quantize audio signals into compact sequences of discrete tokens, and perform the diffusion process entirely in this categorical space. This formulation allows the model to capture high-level semantic structures in audio, while avoiding the challenges of modeling raw waveforms or spectrograms directly. To address these challenges, we introduce a discrete diffusion framework for audio inpainting. Our contributions are threefold, advancing audio inpainting with discrete diffusion model:

- **Discrete diffusion for music.** To the best of our knowledge, our proposed method AIDD, is the first to apply discrete diffusion to tokenized music representations, leveraging the WavTokenizer to transform raw waveforms into compact token sequences. This enables stable generation and long-range semantic coherence.

- **Span-based masking with derivative-regularized reconstruction loss.** We introduce a unified corruption-reconstruction framework that combines structured span masking with a smoothness-oriented regularization loss. The forward process masks contiguous spans of tokens, enabling a progression from fine-grained local corruption to broader semantic perturbations. Additionally, during reconstruction, a derivative-based loss encourages continuity across tokens, penalizing irregular local variations and yielding more natural and perceptually faithful inpainting.

- **Performance.** We present comparable or state-of-the-art results for audio inpainting on MusicNet (Thickstun et al., 2016) and MAESTRO (Hawthorne et al., 2019) datasets, for gaps ranging from 150 - 750 ms, for audio segments containing either a single gap or multiple gaps. We show consistent results for both large and small datasets in both objective and subjective evaluations.

## 2 RELATED WORK

### 2.1 METHODS FOR AUDIO INPAINTING

The term "audio inpainting" was introduced in (Le Roux et al., 2008) to describe the restoration of missing or corrupted audio segments, although related notions had appeared under names such as *audio interpolation* (Marafioti et al., 2020), *audio extrapolation* (Lieb & Stark, 2018), *missing sample reconstruction*, *waveform substitution* (Adler et al., 2011), and *imputation* (Lieb & Stark, 2018).

Classical methods mainly targeted short gaps ($< 100$ ms), often assuming local stationarity. Auto-regressive models (Adler et al., 2011) predicted missing samples from past ones but degrade on longer gaps. Sparse reconstruction in time–frequency domains like STFT or Gabor (Lieb & Stark, 2018; Adler et al., 2011) exploited algorithms such as OMP (Tropp & Gilbert, 2007), later improved by adaptive dictionaries (e.g., learned Gabor atoms (Tauböck et al., 2020)). NMF-based methods (Mokrý et al., 2023) filled spectrogram gaps via low-rank factorization, with probabilistic variants improving robustness. Other structured priors include sinusoidal modeling for harmonic signals (Adler et al., 2011) and graph-based regularization: (Perraudin et al., 2018) used a graph Laplacian on self-similarity matrices to borrow content from uncorrupted regions.

While effective for very short gaps, these models fail for longer ones due to violated assumptions of stationarity or sparsity. Deep learning approaches were introduced to overcome this challenge. CNNs first inpainted spectrograms, with (Marafioti et al., 2019) using a U-Net style context encoder for tens-of-milliseconds gaps. GANs further improved realism: (Ebner & Eltelt, 2020) proposed

---

[1]https://github.com/iftachShoham/AIDD

a Wasserstein GAN with multi-context conditioning for gaps up to 500 ms, while (Marafioti et al., 2020) extended this with GACELA, a multi-scale GAN handling 1–1.5 s gaps. These data-driven models substantially outperform classical approaches, especially on long gaps with complex temporal or musical structure. More recently, A score-aware GAN framework (Aironi et al., 2023a) further improved music inpainting by incorporating symbolic losses, outperforming GACELA on medium gaps.

## 2.2 Audio Inpainting using Diffusion Models

Diffusion models, or score-based generative models, have recently transformed vision and audio generation. They define a forward noising process and learn its reversal (e.g., DDPMs (Ho et al., 2020), score-SDEs (Song et al., 2020)). In audio, they support waveform and spectrogram generation conditioned on context. For instance, DiffWave (Kong et al., 2020) synthesizes raw waveforms for vocoding, while other works apply diffusion in latent or time–frequency domains (STFT, CQT) to exploit structure (Moliner et al., 2023; Moliner & Välimäki, 2023). Most use continuous diffusion, though discrete approaches exist: DiffSound (Yang et al., 2023) applies discrete diffusion to quantized spectrogram tokens. Conceptually, continuous diffusion perturbs real values, yielding smooth interpolations, whereas discrete diffusion masks/replaces categorical tokens, enabling multimodal reconstructions but with possible quantization artifacts (Ho et al., 2020; Song et al., 2020; Yang et al., 2023).

Diffusion has recently been adapted to inpainting. (Moliner & Välimäki, 2023) propose a zero-shot unconditional model on CQT spectrograms, with posterior sampling to enforce data consistency (Kawar et al., 2022; Chung et al., 2022). This handles arbitrary gaps, matching baselines on short ones and surpassing them up to 300 ms. Similarly, CQT-Diff (Moliner et al., 2023) uses an invertible CQT front-end for inpainting, declipping, and bandwidth extension, achieving state-of-the-art restoration. MAID (Liu et al., 2023), instead, trains a conditional diffusion model for music inpainting guided by context and auxiliary cues (e.g., pitch), enabling longer gaps but requiring task-specific supervision. (Aironi et al., 2023a) further propose score-aware conditioning, aligning the score function with masked regions for musically coherent results.

Architectures differ in operating domain (raw waveform vs. spectrogram/latent space) and conditioning. Zero-shot approaches (Moliner & Välimäki, 2023) enforce context consistency during inference, while conditional models integrate masked audio or side information during training. Masking usually involves contiguous spans: Moliner et al. test gaps from 25-300 ms, while GAN baselines address seconds-long gaps (Marafioti et al., 2020; Ebner & Eltelt, 2020). Diffusion models degrade gracefully with gap length: all interpolate short gaps well, but diffusion's generative prior yields more plausible completions beyond 100 ms compared to sparsity- or AR-based methods.

In summary, diffusion-based inpainting combines strong generative priors with flexible conditioning. Open questions remain on continuous vs. discrete diffusion (Yang et al., 2023) and conditioning strategies (posterior sampling vs. trained conditional networks).

## 3 Preliminary

### 3.1 Discrete Diffusion Models

Continuous Diffusion Models (CDMs) (Ho et al., 2020; Song et al., 2020) have achieved remarkable success across various generative modeling tasks, particularly in the image domain (Dhariwal & Nichol, 2021). The high-dimensional nature of raw audio data adds to the difficulty of directly modeling it within the diffusion framework. An effective strategy to address this issue is to first compress continuous audio signals into compact, discrete representations, such as quantized tokens derived from a learned codebook (Défossez et al., 2022). DDMs, which operate in a token space, have recently demonstrated strong performance in the domain of natural language generation (Lou et al., 2024; Nie et al., 2025; Sahoo et al., 2024). Similarly to CDM, DDMs comprise a forward noising process that progressively corrupts an initial sample $x_0$ into a maximally noisy (fully masked) version $x_T$, and a reverse denoising process trained to reconstruct $x_0$ from $x_T$.

**Forward process.** Given the tokenized audio data $x_0 \sim p_{\text{data}}$ over a finite alphabet $\mathcal{X} = \{1, \ldots, N\}$, the forward process is defined by a transition matrix $Q_t$ that indicates how $x_{t-1}$ transits to $x_t$ for each step in the forward process. The behavior of the process is given by:

$$\mathbb{P}(x_{t+\Delta t} = y \mid x_t = x) = \delta_{xy} + Q_t(y, x)\Delta t + \mathcal{O}(\Delta t^2), \tag{1}$$

where $Q_t(y, x)$ denotes the rate of transitioning from state $x$ to state $y$ at time $t$.

Existing work (Lou et al., 2024) shows that the forward diffusion in discrete space can be formulated using a token-level transition rate matrix $Q_{\text{tok}} \in \mathbb{R}^{n \times n}$, where $n$ is the vocabulary size. The forward transition probability of a single token $x_0$ at time $t$ is then given by

$$p_{t|0}^{\text{tok}}(\cdot \mid x_0) = \exp\big(\bar{\sigma}(t)\, Q_{\text{tok}}\big), \tag{2}$$

where $\bar{\sigma}(t) = \int_0^t \sigma(s)\, ds$ denotes the cumulative noise and $Q_{\text{tok}} \in \mathbb{R}^{n \times n}$ is a fixed matrix of absorbing design, where all tokens eventually transition into a terminal [MASK] token.

When $Q_{\text{tok}}$ is set to the absorbing mask transition matrix, the process reduces to only two possible outcomes: the token stays unchanged with probability $e^{-\bar{\sigma}(t)}$, or it is replaced by the [MASK] token with probability $1 - e^{-\bar{\sigma}(t)}$.

**Reverse Process.** The reverse process is described by a new reverse transition matrix (Sun et al., 2022; Kelly, 1981) which is defined as $\bar{Q}_t(y, x) = \frac{p_t(y)}{p_t(x)} Q_t(x, y)$ and $\bar{Q}_t(x, x) = -\sum_{y \neq x} \bar{Q}_t(y, x)$.

To simulate the reverse process, it is sufficient to estimate the ratio $\frac{p_t(y)}{p_t(x)}$, referred to as the *concrete-score* (Lou et al., 2024; Meng et al., 2022), which can be approximated by a neural score function: $s_\theta(x, t) \approx \left[ \frac{p_t(y)}{p_t(x)} \right]_{y \in \mathcal{X},\, y \neq x}$.

**Training.** To train this score function, recent work has proposed the Diffusion Weighted Denoising Score Entropy (DWDSE) objective (Lou et al., 2024), which is defined as:

$$\mathcal{L}_{\text{DWDSE}} = \int_0^T \mathbb{E}_{x_t \sim p_{t|0}(\cdot|x_0)} \left[ \sum_{y \neq x_t} Q_t(x_t, y) \left( s_\theta(x_t, t)_y - \frac{p_{t|0}(y \mid x_0)}{p_{t|0}(x_t \mid x_0)} \log s_\theta(x_t, t)_y + C \right) \right] dt, \tag{3}$$

where $C = K\left( \frac{p_{t|0}(y|x_0)}{p_{t|0}(x_t|x_0)} \right)$, and $K(a) := a \log a - a$ is a convex regularization term. Once trained, we can construct $\mathbf{x}_{t-\Delta t}$ by sampling each token $x_{t-\Delta t}^i$ from the probability

$$p(x_{t-\Delta t}^i \mid x_t^i) = \delta_{x_t^i}(x_{t-\Delta t}^i) + \Delta t\, Q_t^{\text{tok}}(x_t^i, x_{t-\Delta t}^i)\, s_\theta(\mathbf{x}_t, t)_{i,\, x_{t-\Delta t}^i}. \tag{4}$$

In this way, by iteratively applying the reverse step, the model is able to generate new samples.

## 4 AUDIO INPAINTING USING DISCRETE DIFFUSION MODEL

To the best of our knowledge, our method *Audio Inpainting using Discrete Diffusion* (AIDD) is the first to leverage a discrete diffusion model for audio inpainting. AIDD comprises three main components: (1) audio tokenization using WavTokenizer (Ji et al., 2024), (2) generative modeling using a DDM (Lou et al., 2024), and (3) waveform reconstruction via token decoding (Ji et al., 2024). Furthermore, we propose a new training derivative-based loss, and a span-based masking method, extending the approach of (Lou et al., 2024) to more effectively address the audio inpainting task.

### 4.1 THE PROPOSED ARCHITECTURE

**Method Overview.** AIDD consists of WavTokenizer (Ji et al., 2024) that transforms high-dimensional raw audio signals into compact sequences of discrete tokens and reconstructs it back to raw audio, and a Diffusion Transformer (DiT) architecture (Peebles & Xie, 2023) which learns to predict masked tokens through the reverse diffusion process. An overview of the proposed framework is illustrated in Figure 1.

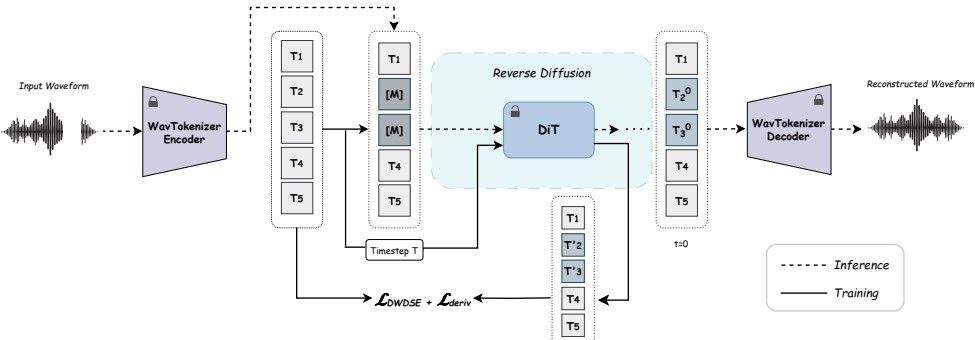

Figure 1: Our method operates on audio signals with missing (silent) segments. During inference, the input waveform, containing a single or multiple silence gaps is processed by the *WavTokenizer* encoder, which converts the audio into a discrete sequence of tokens. Next, a *DiT* performs inpainting by iteratively predicting the masked tokens, resulting in reconstructed token sequence. Finally, the reconstructed tokens are passed through the WavTokenizer's decoder to synthesize the output audio waveform in the masked part. During training, token sequences are corrupted with span-based masking at randomly sampled timesteps, and the DiT is optimized to predict the concrete score using the DWDSE objective, complemented by the derivative-based loss.

**Audio Tokenization.** The first step in our method is to convert the raw audio waveform into discrete audio tokens that can be efficiently processed and trained by the DDM. We utilize a pre-trained audio tokenizer WavTokenizer (Ji et al., 2024), which compresses high-resolution audio into a compact sequence of discrete tokens using a single quantizer. Despite its extreme compression, WavTokenizer preserves high reconstruction fidelity and rich semantic content. Unlike conventional audio inpainting frameworks that operate directly on the audio waveform or the spectrogram (Moliner & Välimäki, 2023), audio tokenization allows us to formulate audio inpainting as a discrete sequence completion task. We also evaluated an alternative tokenizer, UniCodec (Jiang et al., 2025), which is based on a transformer (Vaswani et al., 2017) architecture. Results are presented in Section 7.

**Discrete Diffusion Model.** At the core of our method is a DiT architecture (Peebles & Xie, 2023), which integrates time conditioning into a standard encoder-only transformer (Vaswani et al., 2017; Devlin et al., 2019) and, following the work of (Lou et al., 2024), we incorporate rotary positional encoding (Su et al., 2024). To operate effectively in the discrete token space, we use the denoising score entropy formulation, guiding the reverse diffusion dynamics as seen in Eq. 3.

## 4.2 DERIVATIVE-BASED REGULARIZATION LOSS FOR SMOOTH TOKEN PREDICTIONS

While the DWDSE objective in Eq. 3 ensures that the score function $s_\theta$ learns correct transition ratios for masked tokens, it does not explicitly constrain the *temporal consistency* of predicted token embeddings across neighboring positions. To address this, we introduce a derivative-based regularization term that directly extends the formulation of Section 3 by imposing smoothness on token trajectories in embedding space.

Let $e_i \in \mathbb{R}^d$ denote the ground-truth embedding of the token at position $i$, and $\hat{e}_i \in \mathbb{R}^d$ its predicted embedding. We define discrete temporal derivatives for both ground-truth and predicted embeddings. For first-order differences:

$$\Delta^1 e_i = e_{i+1} - e_i, \quad \Delta^1 \hat{e}_i = \hat{e}_{i+1} - \hat{e}_i,$$

and for second-order differences, capturing local curvature:

$$\Delta^2 e_i = e_{i+1} - 2e_i + e_{i-1}, \quad \Delta^2 \hat{e}_i = \hat{e}_{i+1} - 2\hat{e}_i + \hat{e}_{i-1}.$$

We then define a regularization loss that aligns the derivatives of predicted and ground-truth embeddings at masked locations:

$$\mathcal{L}_{\text{deriv}} = \frac{1}{|M|} \sum_{i \in M} \left\| \Delta^k \hat{e}_i - \Delta^k e_i \right\|^2,$$

where $k \in \{1, 2\}$ selects first- or second-order differences, and $M$ is the set of indices involving masked tokens (or their neighbors).

This additional term complements the DWDSE objective by penalizing irregular local fluctuations in predicted embeddings, thereby encouraging the reverse process to reconstruct sequences that respect the inherent smoothness of natural audio token trajectories. The overall training objective thus combines both components:

$$\mathcal{L}_{\text{total}} = \mathcal{L}_{\text{DWDSE}} + \lambda\, \mathcal{L}_{\text{deriv}},$$

where $\lambda > 0$ is a weighting factor that balances the contribution of the derivative-based regularization.

### 4.3 SPAN MASKING

Whereas prior discrete diffusion frameworks (Lou et al., 2024) apply masking independently in the forward process, we propose a structured corruption strategy that enables the model to better handle inpainting. We employ a *span-based masking* strategy for the absorbing diffusion process.

Given the sequence of tokens $X = (x_1, \ldots, x_L)$ of length $L$, we construct a subset of perturbed tokens by iteratively sampling contiguous spans until a predefined budget is met. This iterative procedure may produce multiple disjoint masked spans within a single timestep. The budget corresponds to the expected number of tokens transitioned away from their original state, defined as $B(t) = \left(1 - e^{-\bar{\sigma}(t)}\right) \cdot L$, where $\bar{\sigma}(t)$ denotes the total noise at time $t$, and $1 - e^{-\bar{\sigma}(t)}$ represents the probability of transitioning away from the current state. This formulation ensures that the proportion of perturbed tokens aligns with the transition probability induced by the forward diffusion process.

At each iteration, we sample a span length $\ell \sim \text{Geo}(p_\sigma)$, where $p_\sigma$ controls the bias toward shorter spans and is defined as $p_\sigma = \frac{p_0}{1 + \alpha \sigma}$, with base parameter $p_0$ and scaling factor $\alpha$ determining how the noise level $\sigma$ influences the expected span length. Thus, early timesteps favor short spans, with longer spans becoming more likely later, capped by $\ell_{\max}$. For each sampled span, a starting index is drawn uniformly from $[1, L - \ell]$, and all tokens in the span are masked.

## 5 EXPERIMENTS

### 5.1 OBJECTIVE METRICS

We adopt a combination of metrics for evaluations, capturing both the technical fidelity and perceptual plausibility of the reconstructed audio.

**Fréchet Audio Distance (FAD).** We use *FAD* (Kilgour et al., 2018), which measures the distributional distance between real and generated audio features. It reflects perceptual similarity in attributes such as timbre and texture, serving as an indicator of realism. Our evaluation uses a VGG-based feature extractor.

**Objective Difference Grade (ODG).** Perceptual quality is further assessed with *ODG* from the PEMO-Q model (Huber & Kollmeier, 2006), which simulates auditory processing and maps a Perceptual Similarity Measure (PSM) to scores from 0 (imperceptible) to –4 (very annoying). We also assessed with ODG using PEA-Q. Both were included to match the evaluation methods of related works. As a reference-based metric aligned with listening tests, ODG complements waveform-based measures.

**Log Spectral Distance (LSD).** Following (Moliner & Välimäki, 2023), we compute *LSD* to capture spectral distortions:

$$\text{LSD} = \frac{1}{T}\sum_{t=1}^{T}\sqrt{\frac{1}{K}\sum_{k=1}^{K}\left(\log|X_{t,k}|^2 - \log|\hat{X}_{t,k}|^2\right)^2} \tag{5}$$

where $X_{t,k}$ and $\hat{X}_{t,k}$ are the STFTs of the original and reconstructed signals. We use a window of $K = 2048$ samples and hop length 512.

**Subjective Listening test.** We conduct a mean opinion score (MOS) test to assess perceptual quality. Participants rate randomly ordered audio segments (1-5 scale), covering different gap lengths and anonymized across methods; original segments are included as a reference. Full evaluation details appear in the supplementary material.

## 5.2 EXPERIMENTAL SETUP

**Dataset.** We evaluated our model on two benchmark datasets: *MusicNet* (Thickstun et al., 2016) and *MAESTRO* (Hawthorne et al., 2019). MusicNet contains 330 classical music recordings with aligned instrument and note annotations. MAESTRO includes over 200 hours of aligned piano performances. We used the predefined train/test splits and trained on each dataset independently.

**Training.** During training, raw audio signals are first tokenized and truncated to a fixed length of 300 tokens (approximately 4 seconds). A random timestep is selected, and noise (in the form of token masking) is applied corresponding to the span-based masking corruption. The DDM is then trained to predict the concrete scores from this corrupted input. Through reverse diffusion steps, the model gradually denoises the token sequence, effectively learning to recover clean representations from noisy ones in token space, illustrated in Figure 1.

Training employed the AdamW optimizer (learning rate of $10^{-6}$), batch size 128, and 300 token samples. On MusicNet, the base model with DWDSE loss was trained for 400k steps (two days) on a single NVIDIA A6000 GPU, while other methods were trained for 100k steps. On MAESTRO, training ran for 150k steps (24h) under the same setup. Full hyperparameters are reported in the supplementary, following ablation results (see Section 7).

**Evaluation.** We evaluate the effectiveness of our proposed audio inpainting method, we followed the experimental protocol of (Moliner & Välimäki, 2023) and selected 60 previously unseen music segments from the *MusicNet* test set (Thickstun et al., 2016), each with a duration of 4.17 seconds. For each segment, we introduced four synthetic gaps at fixed, evenly spaced locations. The gap durations varied across experiments, ranging from 150 ms to 300 ms. Since AIDD is stochastic, we generated 10 samples for each segment and reported the averaged score as the final result.

For the *MAESTRO* dataset (Hawthorne et al., 2019), we followed (Aironi et al., 2023b)'s experimental set-up, taking 6 seconds segments from the predefined test set and introducing single gaps in the middle.

These corrupted inputs were then processed using our method AIDD (see Section 5.2), which reconstructed the missing audio to produce complete, inpainted waveforms. We evaluated the reconstructed samples against the corresponding ground truths using the objective metrics described in Section 5.1.

For subjective evaluation, we introduce MOS subjective listening test, giving the same audio segments for different methods and asking people to rate the quality of audio without prior information on the method, gap or file name. We perform this on the *MAESTRO* dataset. (The full setup is in the Supplementary Materials).

**Inference Pipeline.** At inference, the masked waveform is tokenized with the pretrained *Wav-Tokenizer* encoder. The trained DDM then performs reverse diffusion to predict missing tokens conditioned on context. The token sequence is decoded back to waveform, and only the inpainted segment replaces the original gap, avoiding unnecessary re-encoding of intact regions. A short 10 ms crossfade is applied at boundaries to ensure smooth transitions. An overview is shown in Figure 1.

## 6 RESULTS

For the MusicNet dataset, we evaluate our method against three baselines: **LPC**, an extrapolation method using linear predictive coding (Kauppinen & Roth, 2002); **A-SPAIN-L**, a sparsity-based inpainting approach with dictionary learning (Tauböck et al., 2020); and **CQT-Diff+**, a diffusion model with Constant-Q Transform, state-of-the-art for gaps up to 300 ms (Moliner & Välimäki, 2023).

For the MAESTRO dataset (Hawthorne et al., 2019) we evaluated against three baselines as well: **GACELA** (Marafioti et al., 2020): a cGAN that performs long-gap audio inpainting using multiple discriminators at different time scales; **bin2bin** (Aironi et al., 2023a): a spectrogram inpainting model based on a pix2pix-style GAN with STFT losses; **bin2bin-MIDI** (Aironi et al., 2023a): an extension of bin2bin that adds a piano-roll (MIDI) loss to enforce pitch consistency.

Table 1 reports the average performance across varying gap durations on the *MusicNet* (Thickstun et al., 2016) dataset, while Table 2 presents the corresponding results on the *MAESTRO* (Hawthorne et al., 2019) dataset.

| Method | 150 ms | | | 200 ms | | | 250 ms | | | 300 ms | | |
|---|---|---|---|---|---|---|---|---|---|---|---|---|
| | FAD ↓ | LSD ↓ | ODG ↑ | FAD ↓ | LSD ↓ | ODG ↑ | FAD ↓ | LSD ↓ | ODG ↑ | FAD ↓ | LSD ↓ | ODG ↑ |
| Masked | 16.001 | 0.555 | -3.873 | 18.244 | 0.763 | -3.881 | 23.583 | 0.971 | -3.891 | 33.342 | 1.162 | -3.897 |
| LPC | 3.172 | 0.184 | -3.351 | 4.883 | 0.258 | -3.467 | 7.934 | 0.336 | -3.512 | 11.907 | 0.415 | -3.550 |
| A-SPAIN-L | 6.121 | 0.198 | -3.668 | 12.038 | 0.311 | -3.767 | 16.181 | 0.445 | -3.801 | 21.574 | 0.610 | -3.818 |
| CQT-Diff+ | **1.525** | 0.164 | -3.559 | 2.619 | 0.218 | -3.651 | 3.202 | 0.272 | -3.891 | 4.652 | 0.324 | -3.711 |
| **AIDD** | 1.866 | **0.162** | **-3.215** | **2.391** | **0.209** | **-3.250** | **2.438** | **0.260** | **-3.274** | **3.549** | **0.297** | **-3.284** |

Table 1: Performance comparison of methods across different gap lengths using FAD, LSD, and ODG metrics, on MusicNet dataset (Thickstun et al., 2016). Lower is better for FAD/LSD (↓), higher is better for ODG (↑). In **bold** the best score. The AIDD presented is the highest reported scores, as described in section 5.2.

| Method | 375 ms (↑) | 750 ms (↑) |
|---|---|---|
| GACELA | -3.232 ± 0.232 | -3.318 ± 0.202 |
| bin2bin | -2.892 ± 0.510 | -3.039 ± 0.495 |
| bin2bin-MIDI | -2.800 ± 0.491 | -2.976 ± 0.456 |
| **AIDD** | **-2.303 ± 0.692** | **-2.596 ± 1.300** |

Table 2: ODG (PEA-Q) score values for the three compared methods to AIDD, on MAE-STRO (Hawthorne et al., 2019) dataset. In **bold** the best score. Higher is better (↑). The AIDD's result reported is the combined method.

On the *MusicNet* dataset, AIDD outperforms prior approaches across most gap durations. It achieves the lowest FAD for 200–300 ms gaps and consistently improves LSD and ODG compared to baselines. Against the strong diffusion-based CQT-Diff+, AIDD yields substantially lower distortion for medium and long gaps, with a $\sim 25\%$ FAD reduction at 300 ms (3.549 vs. 4.652). This indicates more perceptually coherent reconstructions as gap size increases. For short gaps (150 ms), CQT-Diff+ attains slightly better FAD, but AIDD still delivers superior ODG and LSD.

Finally, it is worth noting that our base model is almost half the size of CQT-Diff+ and required only one-quarter of the training time, underscoring its efficiency (see Section 8 for latency analysis).

On the *MAESTRO* dataset, results follow a similar trend (Table 2). Our method (AIDD) achieves the best ODG score for the 375 ms gap, outperforming all three baselines by a clear margin. This indicates that AIDD can better preserve perceptual quality even in the presence of interruptions.

Taken together, these findings confirm that AIDD is robust across datasets and gap durations: it provides high perceptual quality for medium and long gaps in *MusicNet*, and achieves strong performance on challenging, real-world piano recordings in *MAESTRO*.

Beyond excelling in objective metrics, AIDD continues to outperform baselines in subjective listening assessments, as shown in Table 3.

Another major advantage to AIDD is that out of all the methods we compared to, our method is the only one that does not require fixed size of input audio.

| Method | MOS (↑) |
|--------|---------|
| Original | $4.12 \pm 0.96$ |
| GACELA | $3.51 \pm 1.33$ |
| CQT-Diff+ | $3.51 \pm 1.34$ |
| **AIDD (WavTokenizer 24kHz)** | **$3.64 \pm 1.26$** |

Table 3: MOS values for the compared methods to AIDD, on MAESTRO (Hawthorne et al., 2019) dataset. In **bold** the best score. Higher is better (↑). The AIDD's result reported is the combined method.

# 7 ABLATION STUDY

To further assess the effectiveness of AIDD, we conducted an ablation study on the MusicNet dataset, following the same experimental protocol described in Section 5.2. Specifically, we examined the impact of different training strategies: *Derivative-Based* (Section 4.2), *Span-Based* (Section 4.3), and their combination. All experiments were performed with the same architecture and inpainting setup to ensure fair comparison. The results are summarized in Table 4.

In addition, we investigated the effect of hyperparameter choices for the proposed training methods. Table 4 reports the performance of each variant, including the *Derivative-Based*, *Span-Based*, and the combined approach.

Overall, the combined method consistently yielded the best performance, achieving the highest scores across most evaluation settings. These results highlight both the robustness of the approach and its alignment with established audio inpainting evaluation protocols.

We also compared between two different tokenizers: UniCodec (Jiang et al., 2025) and WavTokenizer (Ji et al., 2024) on MAESTRO dataset, results shown in Table 5. We also present information loss analysis study between the two in the Supplementary Material. We speculate that although Unicodec pressumed to have better audio quality, the larger codec ($\sim$16k) vs WavTokenizer's ($\sim$4k) may be too large for our model.

| Method / Setting | 200 ms | | | 300 ms | | |
|---|---|---|---|---|---|---|
| | FAD ↓ | LSD ↓ | ODG ↑ | FAD ↓ | LSD ↓ | ODG ↑ |
| AIDD (Base - DWDSE loss) | 2.802 | 0.211 ±0.05 | -3.262 ±0.06 | 4.015 | 0.303 ± 0.06 | -3.296 ±0.04 |
| **AIDD - Span-Based Variants ($\ell_{max} = 30$)** | | | | | | |
| $p_0 = 0.6; \alpha = 0.5$ | 2.438 | **0.206** ±0.05 | **-3.249** ±0.06 | 3.573 | **0.292** ±0.06 | **-3.284** ±0.04 |
| $p_0 = 0.8; \alpha = 0.3$ | 2.719 | 0.211 ±0.05 | -3.258 ±0.06 | 3.626 | 0.298 ±0.07 | -3.289 ±0.04 |
| **AIDD - Derivative-Based loss** | | | | | | |
| $\lambda = 200; \Delta^1 e$ | 2.455 | 0.209 ±0.05 | **-3.251** ±0.06 | **3.439** | **0.293** ±0.06 | **-3.284** ±0.05 |
| $\lambda = 500; \Delta^1 e$ | 2.705 | 0.211 ±0.05 | -3.255 ±0.06 | 3.654 | 0.298 ±0.07 | -3.285 ±0.04 |
| $\lambda = 800; \Delta^1 e$ | 2.869 | 0.215 ±0.05 | -3.260 ±0.06 | 3.837 | 0.303 ±0.07 | -3.289 ±0.04 |
| $\lambda = 200; \Delta^2 e$ | 2.593 | 0.210 ±0.05 | **-3.251** ±0.06 | 3.549 | 0.296 ±0.07 | **-3.281** ±0.05 |
| $\lambda = 500; \Delta^2 e$ | 2.599 | 0.213 ±0.05 | -3.255 ±0.06 | 3.644 | 0.298 ±0.07 | **-3.283** ±0.04 |
| $\lambda = 800; \Delta^2 e$ | 2.539 | 0.213 ±0.05 | -3.256 ±0.06 | 3.538 | 0.301 ±0.07 | **-3.283** ±0.05 |
| **AIDD - Combined methods (Derivative And Span Based) ($\ell_{max} = 30$)** | | | | | | |
| $\lambda = 500 ; p_0 = 0.8; \alpha = 0.5; \Delta^1 e$ | **2.391** | **0.209** ±0.05 | **-3.250** ±0.06 | 3.549 | 0.297 ±0.06 | **-3.284** ±0.04 |
| $\lambda = 200 ; p_0 = 0.8; \alpha = 0.5; \Delta^1 e$ | 2.518 | **0.209** ±0.05 | -3.254 ±0.06 | 3.558 | 0.295 ±0.07 | -3.285 ±0.05 |
| $\lambda = 500 ; p_0 = 0.8; \alpha = 0.5; \Delta^2 e$ | 2.449 | 0.212 ±0.05 | -3.255 ±0.06 | 3.488 | 0.302 ±0.07 | -3.287 ±0.04 |

Table 4: Performance comparison of different **AIDD methods** with WavTokenizer, across different gap lengths using FAD, LSD, and ODG metrics, on MusicNet dataset. In **bold** the best scores. Lower is better for FAD/LSD (↓), higher is better for ODG (↑).

| Model | FAD(↓) | | LSD(↓) | | ODG (↑) | |
|---|---|---|---|---|---|---|
| | 375ms | 750ms | 375ms | 750ms | 375ms | 750ms |
| AIDD (**WavTokenizer 24kHz**) | **0.042** | **0.055** | **0.044** | **0.085** | **-2.303** | **-2.596** |
| AIDD (UniCodec 24kHz) | 0.12 | 0.169 | 0.049 | 0.094 | -2.753 | -3.289 |

Table 5: FAD, LSD, and ODG (PEAQ) scores across gap sizes for the MAESTRO dataset.

## 8 DISCUSSION

**Latency Analysis.** To further evaluate our method AIDD against CQT Diff+, we compared inference performance and model size. All experiments were conducted on a single NVIDIA A6000 GPU using the MAESTRO test set. Because inference time depends on the number of denoising steps rather than the gap length, we report the average time over both 375 ms and 750 ms gaps.

As shown in Table 6, AIDD (WavTokenizer) achieves the fastest inference time while also being the smallest model and fastest to train.

| Model | Param size | Training time | Avg Inference Time (s) | Denoising steps |
|---|---|---|---|---|
| AIDD (WavTokenizer) | **90M (81M)** | **1 day** | **5.25** | 1024 |
| AIDD (UniCodec) | **90M (210M)** | **1 day** | 11.53 | 1024 |
| CQT Diff+ | 242M | 4 days | 12.54 | 35 |

Table 6: Latency comparison of audio-generation, codec type and inverse-model systems.

**Limitation.** AIDD has several limitations. First, its performance is constrained by the quality and bandwidth of the underlying tokenizer; among the two single-codebook codecs we tested, WavTokenizer produced smoother transitions and higher fidelity, but reliance on any specific tokenizer still caps the achievable upper bound (see Supplementary Materials).

Second, comparisons with prior work are affected by cross-domain differences: AIDD uses discrete tokens, whereas baselines generate waveforms or spectrograms. Even with recommended settings, mismatches in reconstruction bandwidth and preprocessing may introduce bias, underscoring the need for a unified benchmark spanning token, latent, and continuous models.

Finally, AIDD inherits WavTokenizer's 24 kHz bandwidth limit, requiring downsampling of higher-rate recordings and restricting outputs to 24 kHz, which may reduce fidelity, unlike other methods that may not subject to that.

**Inference Training Gap.** The model may face a training-inference mismatch: during training, the tokenizer sees a fully intact audio signal before masking (tokenize-then-mask), whereas at inference it must tokenize audio that already contains extended gaps (mask-then-tokenize). Although not fixable, We try to quantify this in the Supplementary Material.

## 9 CONCLUSION

We introduced AIDD, a discrete diffusion framework for audio inpainting that operates in a tokenized space using WavTokenizer and a Diffusion Transformer. By modeling discrete tokens rather than raw waveforms, our approach enables stable training and semantically coherent reconstruction, particularly for long gaps. We further proposed span-based masking and a derivative-based regularization loss, which improve structured corruption modeling and temporal smoothness during reconstruction.

Experiments on MusicNet and MAESTRO demonstrate competitive or superior performance, especially for challenging gaps up to 750 ms. Beyond inpainting, our framework establishes a general paradigm for token-based generative modeling in audio. Future work may explore extending these training strategies to other token-based domains, including language models.

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

## SUPPLEMENTARY MATERIAL

## A    INFORMATION LOSS ANALYSIS

To assess information loss in WavTokenizer and UniCodec, we evaluate FAD and LSD in two stages. First, we measure loss introduced by the tokenizer by encoding and decoding the original audio and comparing it to the reference. Second, we evaluate inpainting by replacing a 750 ms segment with one treated as if produced by the denoising stage, then stitching it using our standard method.

Results show that reconstruction error in the inpainted case is minimal. This indicates that most quality degradation comes from tokenization, not from the denoising process, highlighting the effectiveness of our method while reflecting its limitations.

| Model | FAD($\downarrow$) | | LSD($\downarrow$) | |
|---|---|---|---|---|
| | Tokenized | Inpainted (750ms) | Tokenized | Inpainted (750ms) |
| UniCodec (24kHz) | 2.604 | 0.078 | 0.498 | 0.0645 |
| WavTokenizer (24kHz) | 1.06 | 0.061 | 0.472 | 0.044 |

Table 7: FAD and LSD scores across gap sizes for the MAESTRO dataset for tokenization information loss.

## B    REPRODUCIBILITY STATEMENT

With the following hyperparameters (see Table 8) as well as appropriate hardware, anyone can reproducibly conduct our experiment using our code.

| Parameter | Value |
|---|---|
| Tokens (Unicodec / WavTokenizer) | 16384 / 4096 |
| Learning rate | 1e-6 |
| Batch size | 128 |
| Gradient accumulation | 1 |
| Training steps (MusicNet / MAESTRO) | 100,000 / 150,000 |
| Optimizer | AdamW |
| EMA | 0.9999 |
| Derivative regularization | first_order |
| $\lambda$ | 500 |
| $p_0$ | 0.8 |
| $\alpha$ | 0.5 |
| Temp | 1.0 |
| top-k | 1.0 |
| Span training | True |
| Graph type | absorb |
| Noise schedule | loglinear |
| Predictor | Euler |
| Sampling steps | 128 |

Table 8: Hyperparameters used in experiments.

### B.1    TRAINING ENVIRONMENT

We used singular NVIDIA A6000 GPU for all our experiments. The highest score presented was trained for just less than one day, for 100,000 steps for MusicNet dataset and for the MAESTRO dataset, it was trained for a day, with 150,000 steps.

## C    INFERENCE TRAINING GAP QUANTIZATION

To further examine this point, we conducted an additional experiment to isolate the effect of masking order at inference. Instead of masking the audio waveform before tokenization (mask-then-tokenize), we first tokenized the clean ground-truth audio and then applied masking directly at the token level, so that the tokenizer operated on uninterrupted audio, matching the training setup. The model then performed inpainting on these masked tokens. The results show that the two procedures achieve very similar performance, with only negligible differences, as shown in Table 9.

| Model | FAD(↓) | | LSD(↓) | |
|---|---|---|---|---|
| | 375ms | 750ms | 375ms | 750ms |
| AIDD (mask-then-tokenize) | 0.042 | 0.055 | 0.044 | 0.085 |
| AIDD (tokenize-then-mask) | 0.033 | 0.056 | 0.040 | 0.082 |

Table 9: FAD and LSD scores across gap sizes for the MAESTRO dataset for inference gap analysis.

## D    SUBJECTIVE EVALUATION SETUP

To assess the perceived audio quality of our method in comparison with competing approaches, we conducted a structured subjective evaluation. The goal of this evaluation was to measure how listeners rate the reconstructed audio when gaps of varying lengths are introduced, and to determine whether our method provides a noticeable improvement in perceptual quality.

We began by selecting the same audio clips used in our objective evaluation, all taken from the MAESTRO dataset test set. For each clip, we inserted a single artificial gap of either 375 ms or 750 ms, matching the conditions used in our quantitative experiments. This ensured consistency between the objective and subjective evaluations and enabled a meaningful comparison across methods.

From the full set of processed audio, we randomly chose five samples for each gap duration (a total of ten samples per method). In addition, we included five unmodified audio segments from the dataset to serve as reference samples representing "perfect quality." Before presenting the clips to listeners, we anonymized all files by removing any identifying information about the method used for reconstruction. The order of the clips was also randomized to avoid bias and to ensure that participants could not infer patterns.

All selected audio segments were embedded into a Google Forms questionnaire. Participants were instructed to listen to each segment carefully, using headphones if possible, and to rate its overall quality on a 1–5 Likert scale. A score of 5 corresponded to "excellent quality, indistinguishable from the original," whereas a score of 1 indicated "very poor quality, noticeable artifacts or distortions."

Once the responses were collected, we aggregated the ratings for each method and each gap length. The results of this analysis, including average scores and comparative rankings, are presented and discussed in the main body of the paper.

We gave the following instructions: *You are requested to listen carefully and for each audio rate 1-5 the audio quality in your opinion. 1 being the lowest - very poor quality and 5 being the highest - very high quality. We recommend that you do it on your computer with headsets. Please click on the link, listen, and provide a score based on your opinion.*

## E    ODG IMPLEMENTATIONS: PEMO-Q VS. PEA-Q

In this work we employed two variants of the Objective Difference Grade (ODG): PEMO-Q and PEA-Q. While both provide perceptual quality estimates on the standardized scale from 0 (imperceptible) to -4 (very annoying), they are based on different models and implementations.

**PEMO-Q.** PEMO-Q (Huber & Kollmeier, 2006) is based on a detailed computational model of the auditory system, simulating peripheral and central auditory processing. The ODG values are obtained by mapping a Perceptual Similarity Measure (PSM) to the ITU-defined scale.

**PEA-Q.** PEA-Q follows the ITU-R Recommendation BS.1387 (1999) (Kabal et al., 2002) for objective audio quality assessment, which was designed as a standardized benchmark for audio codec evaluation. In our experiments, we used the MATLAB implementation.

These implementations allowed us to ensure consistency with prior work and comparability across different evaluation protocols.

## F   LLM USAGE STATEMENT

We employed ChatGPT-5 (OpenAI) to refine language and improve clarity of expression in our manuscript. All technical content, experiments, and the final text were conceived, written, and verified solely by the authors.

