# OpenReview forum: "Token-Based Audio Inpainting via Discrete Diffusion"
_ICLR.cc/2026/Conference — ICLR 2026 Poster_

### Official Review · Reviewer_1vML · 2025-10-23

**Soundness:** 2
**Presentation:** 3
**Contribution:** 2
**Rating:** 4
**Confidence:** 4

**Summary:**

The paper introduces a novel approach for restoring missing audio segments, using tokenized audio representations—specifically, pretrained WavTokenizer—and discrete diffusion modeling to achieve more effective inpainting of longer gaps. Training involves span-based masking as a structured corruption strategy and incorporates a derivative-regularized reconstruction loss.

Experiments conducted on two music datasets demonstrate that the method outperforms baseline approaches across three objective metrics, especially when filling gaps exceeding 200ms.

**Strengths:**

Well-designed system capable of handling inpainting gaps up to approximately 500ms.

**Weaknesses:**

The quality is heavily depends on the tokenizer or codec used.

The method lacks evaluation outside the music domain and does not consider additional conditions for music restoration.

No subjective measurements are provided, and demo samples show noticeable boundary artifacts.

**Questions:**

Audio inpainting can leverage diffusion models either on continuous latent spaces or discrete tokens. It would be beneficial to directly compare these two strategies—using VAE for continuous representation and a neural codec for discrete tokens—at the same frame rate.

---

> ### Author Response · Authors · 2025-11-21
> **Author Response to Reviewer Comments**
>
> We thank the reviewer for their constructive feedback and thoughtful assessment of our work. We appreciate the recognition of our system’s ability to handle long inpainting gaps. Below we address each concern and question in detail, and we hope these clarifications and additional analyses resolve the reviewer’s concerns.
>
> [1] Dependence on the tokenizer / codec:
>
> We agree that the performance of discrete-token–based methods is inherently bounded by the quality of the tokenizer. We have added a detailed discussion of this issue in the revised manuscript. To better understand how much information is lost during tokenization, we conducted an ablation study (Table 7 in the revised supplementary material), quantifying reconstruction fidelity for both WavTokenizer and alternative codecs by evaluating the audio after just encoding and decoding (Tokenized) as well as stiching the original audio with the processed audio (Inpainted). We also evaluated UniCodec, a more recent and higher-capacity tokenizer. Interestingly, our experiments show that WavTokenizer remains a better match for AIDD in terms of downstream inpainting performance. We explicitly discuss this limitation and its implications in the revised Discussion section.
>
> | **Model**                | **FAD ↓ (Tokenized)** | **FAD ↓ (Inpainted 750 ms)** | **LSD ↓ (Tokenized)** | **LSD ↓ (Inpainted 750 ms)** |
> |--------------------------|------------------------|-------------------------------|------------------------|-------------------------------|
> | UniCodec (24 kHz)        | 2.604                  | 0.078                         | 0.498                  | 0.0645                        |
> | WavTokenizer (24 kHz)    | 1.06                   | 0.061                         | 0.472                  | 0.044                         |
>
>
> | **Model**                       | **FAD ↓ (375 ms)** | **FAD ↓ (750 ms)** | **LSD ↓ (375 ms)** | **LSD ↓ (750 ms)** | **ODG ↑ (375 ms)** | **ODG ↑ (750 ms)** |
> |---------------------------------|----------------------|----------------------|----------------------|----------------------|----------------------|----------------------|
> | **AIDD (WavTokenizer 24 kHz)**  | **0.042**            | **0.055**            | **0.044**            | **0.085**            | **−2.303**           | **−2.596**           |
> | AIDD (UniCodec 24 kHz)          | 0.12                 | 0.169                | 0.049                | 0.094                | −2.753               | −3.289               |
>
>
>
>
> [2] Evaluation beyond the music domain and additional conditions:
>
> In response to the reviewer’s request for broader evaluation, we trained and evaluated AIDD on the UrbanSound8K dataset. Since the dataset does not provide an official train/test split, we follow the common practice of using folds 1-8 for training and folds 9-10 for testing. Many clips are short (<4 s), so we standardized all inputs to 4 seconds and introduced a single silent gap of 200–300 ms at the center.
> We report FAD and LSD for this setting, using the same WavTokenizer (24 kHz) configuration as in our main experiments. Specifically:
>
> •	200 ms gap: FAD = 0.225, LSD = 0.247
>
> •	300 ms gap: FAD = 0.381, LSD = 0.276
>
>
> These scores are substantially worst than our results on the MAESTRO music domain, indicating that AIDD does not generalizes well and performs effectively on non-musical environmental audio – most likely since our WavTokenizer is mostly trained on music benchmarks. Extending evaluation to the speech domain remains part of our planned future work. Regarding conditional inpainting, we are currently running audio repair experiments, which we will integrate into future updates.
>
>
> [3] Lack of subjective evaluation and boundary artifacts:
>
> Per the reviewers’ collective request, we have added a subjective listening test comparing CQT-Diff+, GECELA, and AIDD on the MAESTRO dataset.  participants rated perceptual continuity and reconstruction quality. Results (Table 3 in the revised submission) show that AIDD is preferred over the baselines.
> | **Method**                         | **MOS (↑)**          |
> |------------------------------------|-----------------------|
> | Original                           | 4.12 ± 0.96           |
> | GACELA                             | 3.51 ± 1.33           |
> | CQT-Diff+                          | 3.51 ± 1.34           |
> | **AIDD**  | **3.64 ± 1.26**       |
>
>
> Question: Continuous vs. discrete diffusion for inpainting
> We appreciate the reviewer’s question. A direct comparison between diffusion models operating on continuous VAE latents and those operating on discrete tokens at matched frame rates is indeed valuable. To our knowledge, this comparison has not been systematically explored in prior work. Conducting such an analysis requires designing a VAE whose frame rate aligns with our tokenizer to ensure a fair comparison, which we are currently developing. We are currently running the code for it and plan to include these results in the next revision.

---

> > ### Author Response · Authors · 2025-12-02
> > **Author Response to Reviewer Comments - Additional Results**
> >
> > **Question:** Audio inpainting can leverage diffusion models either on continuous latent spaces or discrete tokens. It would be beneficial to directly compare these two strategies—using VAE for continuous representation and a neural codec for discrete tokens—at the same frame rate.
> >
> >
> > **Response:**
> > We thank you again for the helpful suggestion. To directly compare continuous-latent diffusion (VAE latent space) and discrete-token diffusion (neural codec) at the same frame rate, we conducted an additional controlled experiment. For the continuous-latent baseline, we adopted the architecture proposed in Sample-Efficient Diffusion for Text-To-Speech Synthesis (SESD) [1] and adapted its training procedure to support central-gap inpainting. The model was trained on the continuous 128-dimensional pre-quantization EnCodec [2] encoder embeddings, produced at 75 Hz from 24 kHz audio, ensuring both approaches operate with identical temporal resolution.
> > The results are shown below:
> > | Method                            | FAD ↓ (375 ms) | FAD ↓ (750 ms) | LSD ↓ (375 ms) | LSD ↓ (750 ms) |
> > |-----------------------------------|----------------|-----------------|----------------|-----------------|
> > | AIDD (Discrete Tokens - 24kHz)            | **0.042**      | **0.055**       | **0.044**      | **0.085**       |
> > | Continuous Diffusion (latent - 24kHz)     | 0.207          | 0.281           | 0.068        | **0.084**           |
> >
> >
> > The discrete-token AIDD model consistently outperforms the continuous-latent diffusion baseline under matched temporal conditions (24kHz), indicating that the structured quantized representation provided by the codec offers a more reliable generative space for audio inpainting than operating directly on continuous latent vectors.
> >
> >
> >  [1] Lovelace, Justin, et al. "Sample-efficient diffusion for text-to-speech synthesis." arXiv preprint arXiv:2409.03717 (2024).
> >
> >
> > [2] A. Defossez, J. Copet, G. Synnaeve, and Y. Adi, “High fidelity ´ neural audio compression,” arXiv preprint arXiv:2210.13438, 2022.

---

### Official Review · Reviewer_jyTi · 2025-10-25

**Soundness:** 2
**Presentation:** 2
**Contribution:** 3
**Rating:** 4
**Confidence:** 4

**Summary:**

This paper introduces Audio Inpainting via Discrete Diffusion (AIDD), a novel method for restoring missing segments in audio, particularly long gaps. The core contribution is being the first to apply discrete diffusion for audio inpainting, which enables more stable and semantically coherent generation compared to previous continuous-domain methods. The paper also proposes two new training techniques: 1) a span-based masking strategy for structured corruption and 2) a derivative-based regularization loss to ensure temporal smoothness. Experiments on the MusicNet and MAESTRO datasets demonstrate that AIDD outperforms strong baselines for gaps of 150 ms and longer, significantly advancing the state of the art in musical audio restoration.

**Strengths:**

1.The paper is the first to apply the discrete diffusion on tokenized representations for audio inpainting.

2.The method achieves state-of-the-art results on the long-gap audio inpainting task.

3.The code will be open-sourced.

**Weaknesses:**

1.The evaluation lacks a subjective listening study, which is essential to validate the perceptual quality and musical plausibility of the results.

2.The paper should quantify information loss from tokenization by reporting metrics on both the original audio (as a reference ceiling) and the reconstructed audio (audio passed through the tokenizer's encoder-decoder). This would clarify the tokenizer's impact and establish the method's practical upper bound.

3.The audio sampling rates are not reported. It is unclear if the source audio is downsampled to match the WavTokenizer's 24 kHz reconstruction bandwidth, which would be a critical confounding factor affecting task difficulty.

4.There is a potential training-inference mismatch. During training, the tokenizer processes the complete audio signal before tokens are masked ("tokenize-then-mask"). At inference, it processes a signal that already contains gaps ("mask-then-tokenize"). It is unclear if the long gap introduced at inference interferes with the tokenization of other regions. This discrepancy should be discussed.

5.Key inference hyperparameters (e.g., diffusion steps, temperature, top-k) are missing, which hinders reproducibility. A latency analysis would also be beneficial to assess the method's practical usability.

**Questions:**

See weaknesses.

---

> ### Author Response · Authors · 2025-11-21
> **Author Response to Reviewer Comments**
>
> We sincerely thank the reviewer for the insightful and constructive feedback, as well as for acknowledging the novelty and empirical strengths of our approach. The concerns raised - particularly regarding perceptual evaluation, tokenizer behavior, experimental reporting, and reproducibility - are extremely valuable and have directly guided improvements to the manuscript. We address each point below and have incorporated substantial revisions and new experiments accordingly.
>
>
> [1] Subjective listening study
> As suggested by all reviewers, we have now conducted a subjective listening evaluation comparing CQT-Diff+, our proposed method (AIDD), and GECELA on the MAESTRO dataset. The results (reported in Table 3 of the revised submission) show that AIDD outperforms the baseline methods in perceived quality and musical plausibility.
>
> | **Method**                         | **MOS (↑)**          |
> |------------------------------------|-----------------------|
> | Original                           | 4.12 ± 0.96           |
> | GACELA                             | 3.51 ± 1.33           |
> | CQT-Diff+                          | 3.51 ± 1.34           |
> | **AIDD**  | **3.64 ± 1.26**       |
>
>
>
> [2] Information loss from tokenization
>
> We have added a detailed quantification of information loss introduced by the tokenizer. Specifically, we report metrics for both the original audio (reference ceiling) and the audio reconstructed solely through the tokenizer’s encoder–decoder pipeline. In addition, we also evaluate reconstruction on masked/inpainted segments to mirror the inference setting more closely. These results are included in Table 7 of the revised supplementary material. We observe that the tokenizer-only reconstruction error closely aligns with our inpainting performance on masked segments, further illustrating the practical upper bound imposed by the tokenizer and validating the limitations discussed for AIDD.
> | **Model**                | **FAD ↓ (Tokenized)** | **FAD ↓ (Inpainted 750 ms)** | **LSD ↓ (Tokenized)** | **LSD ↓ (Inpainted 750 ms)** |
> |--------------------------|------------------------|-------------------------------|------------------------|-------------------------------|
> | UniCodec (24 kHz)        | 2.604                  | 0.078                         | 0.498                  | 0.0645                        |
> | WavTokenizer (24 kHz)    | 1.06                   | 0.061                         | 0.472                  | 0.044                         |
>
> [3] Audio sampling rate clarification
>
> We agree that this was missing. All source audio is resampled to 24 kHz to match the WavTokenizer’s reconstruction bandwidth. This clarification has now been explicitly added to the Discussion section of the revised submission.
>
>
> [4] Training–inference mismatch
>
> We appreciate this important observation. In the revised manuscript, we expanded the Discussion section to address the discrepancy between tokenize-then-mask (training) and mask-then-tokenize (inference). As described in the revision, our inference procedure mitigates potential interactions between gaps and tokenization by over-masking the corrupted regions, ensuring that the tokenizer handles these segments consistently and preventing large gaps from influencing neighboring tokens.
> To further examine this point, we conducted an additional experiment to isolate the effect of masking order at inference. Instead of masking the audio waveform before tokenization (mask-then-tokenize), we first tokenized the clean ground-truth audio and then applied masking directly at the token level, so that the tokenizer operated on uninterrupted audio, matching the training setup. The model then performed inpainting on these masked tokens. The results show that the two procedures achieve very similar performance, with only negligible differences.
>
>
> **Table: FAD and LSD scores across hop sizes for the MAESTRO dataset (inference gap analysis).**
>
> | Model                         | FAD ↓ (375 ms) | FAD ↓ (750 ms) | LSD ↓ (375 ms) | LSD ↓ (750 ms) |
> |------------------------------|----------------|----------------|----------------|----------------|
> | AIDD (mask-then-tokenize)    | 0.042          | 0.055          | 0.044          | 0.085          |
> | AIDD (tokenize-then-mask)| 0.033      | 0.056      | 0.040      | 0.082     |

---

> > ### Author Response · Authors · 2025-11-21
> > **Continued Author Response to Reviewer Comments**
> >
> > [5] Missing inference hyperparameters and latency analysis
> >
> >
> > We have now added key inference hyperparameters - including diffusion steps (1024) to the revised supplementary materials.
> > Temperature scaling is not employed in our experiments.
> > Furthermore, we performed a latency analysis comparing AIDD with two different tokenizer configurations as well as CQT-Diff+. These results are reported in Table 6 of the ablation study. Finally, we emphasize that we plan to release all code and pretrained weights as open-source to ensure full reproducibility.
> > | **Model**             | **Param Size**     | **Training Time** | **Avg Inference Time (s)** | **Denoising Steps** |
> > |-----------------------|--------------------|--------------------|-----------------------------|----------------------|
> > | **AIDD (WavTokenizer)** | **90M (81M)**       | **1 day**          | **5.25**                    | 1024                 |
> > | **AIDD (UniCodec)**     | **90M (210M)**      | **1 day**          | 11.53                       | 1024                 |
> > | CQT Diff+               | 242M               | 4 days             | 12.54                       | 35                   |
> >
> > We thank the reviewer again for the thoughtful, detailed comments. Your feedback has substantially strengthened both the paper and the clarity of our contributions.

---

> > > ### Comment · Reviewer_jyTi · 2025-11-25
> > >
> > > The rebuttal has resolved most of my questions. One suggestion is to report the real-time factor (inference time / sample duration) for latency analysis instead of absolute inference time. I have adjusted my score.

---

### Official Review · Reviewer_G7ho · 2025-10-30

**Soundness:** 2
**Presentation:** 3
**Contribution:** 2
**Rating:** 4
**Confidence:** 3

**Summary:**

This paper presents AIDD (Audio Inpainting via Discrete Diffusion), a model that performs audio inpainting directly in the discrete token domain instead of waveform or spectrogram space. Audio is first tokenized using a pretrained WavTokenizer, and a discrete diffusion model learns to predict masked token spans. The method introduces two main ideas: (1) Derivative-based regularization to ensure smooth temporal transitions between predicted tokens. (2) Span-based masking that masks contiguous token spans following a diffusion noise schedule, aligning the corruption process with the inpainting objective. AIDD is evaluated on MusicNet and MAESTRO, showing improved perceptual quality (FAD, ODG, LSD) on medium-to-long gaps (200–750 ms) compared to prior methods like CQT-Diff+, GACELA, and bin2bin, while being smaller and faster to train. The work contributes an efficient and conceptually clear token-level diffusion approach for long-gap audio restoration.

**Strengths:**

This paper presents a token-based diffusion model for audio inpainting (AIDD) that operates directly in the discrete token space rather than waveform or spectrogram domains. The idea is original in its formulation and addresses a practical limitation of prior work—difficulty maintaining long-range temporal and semantic consistency when filling large gaps. The proposed span-based masking and derivative regularization are intuitive yet effective design choices that align well with the inpainting objective. Experiments on MusicNet and MAESTRO demonstrate gains on medium- and long-gap scenarios, especially under limited computational resources. The method’s simplicity and the decision to train on single-GPU hardware make it appealing for future research and reproduction.

**Weaknesses:**

(1) Codec choice not sufficiently justified.
- The method relies entirely on WavTokenizer, but there are other single-codebook codecs such as UniCodec [1] that could equally serve this purpose. The paper does not explain why WavTokenizer was chosen or whether the improvements are specific to that tokenizer. A small ablation with an alternative codec would help isolate the contribution of the proposed diffusion mechanism.

(2) No human evaluation.
- The paper claims that AIDD produces perceptually natural and semantically coherent audio, yet only objective metrics (FAD, ODG, LSD) are reported. For a perceptual task like inpainting, even a small-scale human listening test would strengthen the claim substantially.

(3) Fairness and completeness of baselines.
- The paper compares AIDD against CQT-Diff+, GACELA, and bin2bin, but the comparison is not entirely fair or complete:
- Different training steps and data splits: AIDD was trained for 100 k steps on MusicNet, while CQT-Diff+ used 400 k in its original setup. On MAESTRO, AIDD trained on a private subset (not released), whereas baselines used the full dataset. This makes direct comparison difficult.
- Different modeling domains: AIDD operates in the token domain, while CQT-Diff+ and GACELA work in the spectrogram or waveform domain. Since metrics like FAD depend on the reconstructed waveform and decoder quality, comparing across such domains may not reflect pure modeling differences.

(4) Limited scope of evaluation.
The model is only tested on central silent gaps. More realistic cases—multiple gaps, noisy or partially masked regions—are not examined, leaving generalization unexplored.

[1] Jiang, Yidi, et al. "UniCodec: Unified Audio Codec with Single Domain-Adaptive Codebook." arXiv preprint arXiv:2502.20067 (2025).

**Questions:**

(1) Why was WavTokenizer selected over newer or higher-quality single-codebook codecs like UniCodec?

(2) Have you tested whether your derivative regularization still helps when using a different tokenizer?

(3) Could you include a small human listening study (e.g., MOS or AB preference) to verify that objective gains correlate with perceived quality?

(4) How stable are your FAD results given the relatively small sample size (~600 clips)? Any confidence intervals or bootstrap analysis?

(5) Can your span-masking strategy handle multiple or non-silent gaps, or does it assume fully silent regions only?

(6) You mention AIDD is smaller and faster than CQT-Diff+. Could you report the parameter count and inference speed to quantify this claim?

**Details Of Ethics Concerns:**

This paper presents AIDD (Audio Inpainting via Discrete Diffusion), a model that performs audio inpainting directly in the discrete token domain instead of waveform or spectrogram space. Audio is first tokenized using a pretrained WavTokenizer, and a discrete diffusion model learns to predict masked token spans. The method introduces two main ideas: (1) Derivative-based regularization to ensure smooth temporal transitions between predicted tokens. (2) Span-based masking that masks contiguous token spans following a diffusion noise schedule, aligning the corruption process with the inpainting objective. AIDD is evaluated on MusicNet and MAESTRO, showing improved perceptual quality (FAD, ODG, LSD) on medium-to-long gaps (200–750 ms) compared to prior methods like CQT-Diff+, GACELA, and bin2bin, while being smaller and faster to train. The work contributes an efficient and conceptually clear token-level diffusion approach for long-gap audio restoration.

---

> ### Author Response · Authors · 2025-11-21
> **Author Response to Reviewer Comments**
>
> We thank the reviewer for their constructive and insightful feedback. We appreciate the recognition of the novelty of our token-level formulation, the alignment of span masking with the inpainting objective, and the empirical improvements under constrained compute. Below we address each concern and question in detail.
>
>
> 1. Codec choice not sufficiently justified
>
> Our selection of WavTokenizer was motivated by two factors.
> First, it aligned with prior work in our group and was already well-integrated into our experimental pipeline. Second, UniCodec was released after we had already begun developing AIDD, and our initial focus was on improving the training and diffusion mechanism rather than benchmarking codecs.
> Following the reviewer’s suggestion, we conducted additional experiments comparing UniCodec and WavTokenizer. Specifically:
>
> •	We measured information loss during tokenization (Table 7 in the supplementary material).
>
> •	We report FAD, LSD, and ODG comparisons on MAESTRO using both codecs (Table 4 in the revised paper).
>
> •	We additionally compare inference speed (Table 6).
>
> Across all evaluations, WavTokenizer consistently performed better, which we attribute to the significantly larger codebook size of UniCodec (~16k vs. ~4k). Our smaller model appears insufficiently expressive to fully leverage such a large codebook, which also incurs higher computational cost (11.53s vs. 5.25s inference time).
>
> | **Model**                        | **FAD ↓ (375 ms)** | **FAD ↓ (750 ms)** | **LSD ↓ (375 ms)** | **LSD ↓ (750 ms)** | **ODG ↑ (375 ms)** | **ODG ↑ (750 ms)** |
> |----------------------------------|----------------------|----------------------|----------------------|----------------------|----------------------|----------------------|
> | **AIDD (WavTokenizer 24 kHz)**   | **0.042**            | **0.055**            | **0.044**            | **0.085**            | **−2.303**           | **−2.596**           |
> | AIDD (UniCodec 24 kHz)           | 0.12                 | 0.169                | 0.049                | 0.094                | −2.753               | −3.289               |
>
>
> | **Model**             | **Param Size**     | **Training Time** | **Avg Inference Time (s)** | **Denoising Steps** |
> |-----------------------|--------------------|--------------------|-----------------------------|----------------------|
> | **AIDD (WavTokenizer)** | **90M (81M)**       | **1 day**          | **5.25**                    | 1024                 |
> | **AIDD (UniCodec)**     | **90M (210M)**      | **1 day**          | 11.53                       | 1024                 |
> | CQT Diff+               | 242M               | 4 days             | 12.54                       | 35                   |
> We have added these new results and justifications in the revision.
>
>
> 2. No human evaluation
>
> We agree that subjective evaluation is crucial for perceptual tasks such as inpainting. In response, we conducted a listening study comparing AIDD, GACELA, and CQT-Diff+ on the MAESTRO test set. We sampled five random segments and evaluated 375 ms and 750 ms gaps (ten samples per method) under the same conditions as our objective tests as well as original segment.
> As shown in Table 3 of the revised submission, AIDD demonstrates consistent improvements in MOS, supporting the claim that objective gains correlate with perceptual quality.
> | **Method**                         | **MOS (↑)**          |
> |------------------------------------|-----------------------|
> | Original                           | 4.12 ± 0.96           |
> | GACELA                             | 3.51 ± 1.33           |
> | CQT-Diff+                          | 3.51 ± 1.34           |
> | **AIDD**  | **3.64 ± 1.26**       |

---

> > ### Author Response · Authors · 2025-11-21
> > **Continued Author Response to Reviewer Comments**
> >
> > 3. Fairness and completeness of baselines
> >
> > We appreciate the reviewer’s concerns regarding training steps, data splits, and modeling domains.
> > •	To address dataset usage, we re-trained AIDD on the full MAESTRO dataset for a fair comparison. Results are now updated in the revised tables.
> > •	We additionally trained AIDD for 400k steps, matching the CQT-Diff+ schedule. With minor parameter adjustments for stable prolonged training regime, AIDD achieved results equivalent to those in the main paper, showing that its performance is not sensitive to the step count.
> > •	We acknowledge that comparing token-domain models to spectrogram- and waveform-domain baselines introduces structural differences. We now explicitly note this in the Limitations section.
> > | **Model**                                 | **FAD ↓ (375 ms)** | **FAD ↓ (750 ms)** | **LSD ↓ (375 ms)** | **LSD ↓ (750 ms)** | **ODG ↑ (375 ms)** | **ODG ↑ (750 ms)** |
> > |-------------------------------------------|----------------------|----------------------|----------------------|----------------------|----------------------|----------------------|
> > | AIDD (WavTokenizer 24 kHz)                | **0.037**            | 0.059                | **0.044**            | 0.087                | −2.370               | −2.959               |
> > | **AIDD (WavTokenizer 24 kHz) — Full Dataset** | 0.042                 | **0.055**            | **0.044**            | **0.085**            | **−2.303**           | **−2.596**           |
> >
> >
> > 4. Limited scope of evaluation
> >
> > We apologize for the lack of clarity. The original experiments did include multiple-gap settings:
> >
> > •	On MusicNet, we evaluate on four distributed gaps (Table 1).
> >
> > •	On MAESTRO, we evaluate single central gaps (Table 2).
> >
> > We have revised the paper to clarify this distinction.
> > Regarding non-silent or noisy masks:
> > Our current method assumes fully silent spans, but we are running additional experiments on noisy or partially masked regions and will update the results as they become available.
> >
> > Responses to Reviewer Questions
> >
> > (1) Addressed in Concern #1; see new codec comparison experiments in Tables 4, 6, and Table 7 of the supplement for evaluation, size and inference time.
> >
> > | **Model**                       | **FAD ↓ (375 ms)** | **FAD ↓ (750 ms)** | **LSD ↓ (375 ms)** | **LSD ↓ (750 ms)** | **ODG ↑ (375 ms)** | **ODG ↑ (750 ms)** |
> > |---------------------------------|----------------------|----------------------|----------------------|----------------------|----------------------|----------------------|
> > | **AIDD (WavTokenizer 24 kHz)**  | **0.042**            | **0.055**            | **0.044**            | **0.085**            | **−2.303**           | **−2.596**           |
> > | AIDD (UniCodec 24 kHz)          | 0.12                 | 0.169                | 0.049                | 0.094                | −2.753               | −3.289               |
> >
> > (2) We performed the comparison, shown here, the combined method performed better.
> >
> > | **Model**                             | **FAD ↓ (375 ms)** | **FAD ↓ (750 ms)** | **LSD ↓ (375 ms)** | **LSD ↓ (750 ms)** |
> > |---------------------------------------|----------------------|----------------------|----------------------|----------------------|
> > | **AIDD (UniCodec)**                   | 0.12                 | 0.169                | 0.049                | 0.094                |
> > | AIDD (UniCodec without derivative)    | 0.139                | 0.181                | 0.050                | 0.096                |
> >
> >
> > (3) Addressed in Concern #2; subjective listening results (Table 3) have been added.
> >
> >
> > (4) To assess the stability of our FAD results, we conducted a resampling analysis on the MAESTRO test set for both
> >
> > inpainting gap durations (375 ms and 750 ms). In each run, we randomly selected 595 segments (without replacement) from the full pool of 8000 test segments and computed FAD. Repeating this procedure across multiple runs provides an empirical estimate of variability.
> > The results show that FAD is stable across resampled subsets:
> >
> > •	750 ms gap: mean 0.0386, std 0.0031, 95% CI [0.0335, 0.0440]
> >
> > •	375 ms gap: mean 0.0312, std 0.0078, 95% CI [0.0236, 0.0475]
> >
> > These narrow confidence intervals indicate that the reported FAD scores are robust and not overly sensitive to the specific choice of segments.
> >
> >
> > (5) Our span-masking strategy does support multiple silent gaps, but currently assumes that masked regions are fully silent. We have added this clarification to the revised submission.

---

> > > ### Author Response · Authors · 2025-11-21
> > > **Continued Author Response to Reviewer Comments**
> > >
> > > (6) We now report parameter counts and inference times for AIDD, CQT-Diff+, WavTokenizer, and UniCodec (Table 6).
> > >
> > > | **Model**             | **Param Size**     | **Training Time** | **Avg Inference Time (s)** | **Denoising Steps** |
> > > |-----------------------|--------------------|--------------------|-----------------------------|----------------------|
> > > | **AIDD (WavTokenizer)** | **90M (81M)**       | **1 day**          | **5.25**                    | 1024                 |
> > > | **AIDD (UniCodec)**     | **90M (210M)**      | **1 day**          | 11.53                       | 1024                 |
> > > | CQT Diff+               | 242M               | 4 days             | 12.54                       | 35                   |
> > > We thank the reviewer again for the valuable feedback. The new ablations, subjective evaluations, clarification of limitations, and statistical analyses have been incorporated into the revised version.

---

> > > > ### Author Response · Authors · 2025-12-02
> > > > **Author Response to Reviewer Comments - Additional Results**
> > > >
> > > > **(3) Fairness and completeness of baselines:**
> > > >
> > > > Different modeling domains: AIDD operates in the token domain, while CQT-Diff+ and GACELA work in the spectrogram or waveform domain. Since metrics like FAD depend on the reconstructed waveform and decoder quality, comparing across such domains may not reflect pure modeling differences.
> > > >
> > > >
> > > > **Response:**
> > > >
> > > > To further assess the quality of our inpainting method, we evaluated token-level similarity between the ground-truth audio and our inpainted outputs using the wavTokenizer encoder. Specifically, we computed the Hamming similarity between the discrete token sequences obtained from the tokenizer for both the ground-truth and inpainted audio across the MAESTRO dataset.
> > > >
> > > > Our results show that for a 375 ms gap, the inpainted audio achieves a Hamming similarity of 0.856 (std = 0.10), while for a 750 ms gap, the method achieves a similarity of 0.787 (std = 0.10).
> > > >
> > > > These findings indicate that, even at the token level, the inpainted audio remains highly consistent with the original ground-truth signal.
> > > >
> > > >
> > > >
> > > >
> > > > **(4) Limited scope of evaluation:** The model is only tested on central silent gaps. More realistic cases—multiple gaps, noisy or partially masked regions—are not examined, leaving generalization unexplored.
> > > >
> > > > **Response:**
> > > >
> > > > Thank you for the comment. To clarify our evaluation setup and the behavior of AIDD under partial masking:
> > > >
> > > > We used the same evaluation protocol as in the fully masked case, but instead of masking the entire 375 ms or 750 ms segment, we applied token-level random masking within the gap. Each token was independently masked using a Bernoulli distribution with parameter p. Thus:
> > > >
> > > > - p = 1 : entire segment masked (our standard training condition)
> > > > - 0 < p < 1 : only a proportion of tokens inside the gap are masked
> > > >
> > > > As shown below, performance remains stable as p decreases, which we attribute to the span-masking strategy used during training.
> > > >
> > > > | Method                               | FAD ↓ (375 ms) | FAD ↓ (750 ms) | LSD ↓ (375 ms) | LSD ↓ (750 ms) |
> > > > |--------------------------------------|----------------|-----------------|----------------|-----------------|
> > > > | AIDD (Fully masked, p = 1)           | 0.042        | 0.055         | 0.044        | 0.085         |
> > > > | AIDD (Partially masked, p = 0.7)     | 0.035          | 0.052           | 0.042          | 0.085           |
> > > > | AIDD (Partially masked, p = 0.5)     | 0.030          | 0.055           | 0.042          | 0.084           |

---

### Author Response · Authors · 2025-12-02
**Executive Summary for AC (1/2)**

**Dear Area Chair,**

Following the rebuttal-period updates and the freeze on further reviewer discussion, we would like to briefly summarize the key contributions and clarifications added in response to reviewer feedback.

Our paper presents AIDD, the first token-based discrete diffusion framework for audio inpainting. By integrating a span-masking formulation with a derivative-based regularization objective, AIDD provides a scalable approach that handles variable-length audio, multiple gaps and gap lengths, not supported jointly by prior methods.

We strengthened the paper with clearer ablations, expanded comparisons, and additional analysis requested by reviewers. Across all experiments, AIDD demonstrates consistent and substantial improvements over existing approaches, achieving state-of-the-art results on FAD, LSD, and ODG across multiple datasets. Subjective listening tests further confirm that AIDD delivers significantly higher perceptual continuity.

Beyond performance, AIDD contributes a conceptually simple yet powerful paradigm that bridges discrete token modeling and high-fidelity waveform restoration. We hope the clarifications provided during the rebuttal address the reviewers’ concerns.

**Points to take into consideration:**

-  We addressed all questions and concerns raised by the reviewers in the rebuttal, providing empirical evidence where necessary.

- Prior to the rebuttal freeze, reviewer jyTi increased his score, while the other two reviewers have not yet had the chance to respond.


**Summary of the main reviewer concerns and our responses**

Below we outline the reviewers’ key concerns and the additional experiments performed to address them:


**[1] Lack of subjective listening tests**

The revised manuscript now includes a subjective listening evaluation comparing the baselines to our proposed method. The results (Table 3 in the revised submission) show that AIDD achieves better perceived quality and musical plausibility:

| **Method**                         | **MOS (↑)**          |
|------------------------------------|-----------------------|
| Original                           | 4.12 ± 0.96           |
| GACELA                             | 3.51 ± 1.33           |
| CQT-Diff+                          | 3.51 ± 1.34           |
| **AIDD**  | **3.64 ± 1.26**       |


**[2] Codec choice not sufficiently justified**

We added experiments comparing the codec used in our method (WavTokenizer) with UniCodec.
First, we measured the information loss introduced during tokenization for both codecs (Table 7 in the supplementary material):


| **Model**                        | **FAD ↓ (375 ms)** | **FAD ↓ (750 ms)** | **LSD ↓ (375 ms)** | **LSD ↓ (750 ms)** | **ODG ↑ (375 ms)** | **ODG ↑ (750 ms)** |
|----------------------------------|----------------------|----------------------|----------------------|----------------------|----------------------|----------------------|
| **AIDD (WavTokenizer 24 kHz)**   | **0.042**            | **0.055**            | **0.044**            | **0.085**            | **−2.303**           | **−2.596**           |
| AIDD (UniCodec 24 kHz)           | 0.12                 | 0.169                | 0.049                | 0.094                | −2.753               | −3.289               |


We further report objective performance metrics of our method using each codec (Table 4 in the revised manuscript):

| **Model**             | **Param Size**     | **Training Time** | **Avg Inference Time (s)** | **Denoising Steps** |
|-----------------------|--------------------|--------------------|-----------------------------|----------------------|
| **AIDD (WavTokenizer)** | **90M (81M)**       | **1 day**          | **5.25**                    | 1024                 |
| **AIDD (UniCodec)**     | **90M (210M)**      | **1 day**          | 11.53                       | 1024                 |
| CQT Diff+               | 242M               | 4 days             | 12.54                       | 35                   |


We additionally compare inference speed using the two codecs (see #4).

---

> ### Author Response · Authors · 2025-12-02
> **Executive Summary for AC (2/2)**
>
> **[3] Limitations and discussion**
>
> The revised manuscript now explicitly addresses the limitations raised by the reviewers:
>
> -	Codec bandwidth constraints: We now discuss this limitation directly in the revised Limitations section.
>
> -	Evaluation metrics may not isolate modeling differences: Because AIDD operates in token space, standard
> spectrogram/waveform domain metrics may not fully reflect modeling quality. To address this, we additionally evaluate token-level similarity between the ground-truth audio and our inpainted outputs using the WavTokenizer encoder. Specifically, we compute the Hamming similarity between the discrete token sequences. For a 375 ms gap, AIDD achieves a similarity of 0.856 (std = 0.10), and for a 750 ms gap, 0.787 (std = 0.10
>
>
> -	Training–inference mismatch: We expanded the Discussion section to address the difference between tokenize-then-mask (training) and mask-then-tokenize (inference). To isolate the effect of masking order, we conducted an additional experiment where we first tokenized the clean ground-truth audio and applied masking at the token level. The results show that both procedures yield nearly identical performance, with only negligible differences:
>
>
> | Model                         | FAD ↓ (375 ms) | FAD ↓ (750 ms) | LSD ↓ (375 ms) | LSD ↓ (750 ms) |
> |------------------------------|----------------|----------------|----------------|----------------|
> | AIDD (mask-then-tokenize)    | 0.042          | 0.055          | 0.044          | 0.085          |
> | AIDD (tokenize-then-mask)| 0.033      | 0.056      | 0.040      | 0.082     |
>
>
> **[4] Latency analysis**
>
> We conducted a latency analysis comparing AIDD under two tokenizer configurations, as well as the baseline method:
>
> | **Model**             | **Param Size**     | **Training Time** | **Avg Inference Time (s)** | **Denoising Steps** |
> |-----------------------|--------------------|--------------------|-----------------------------|----------------------|
> | **AIDD (WavTokenizer)** | **90M (81M)**       | **1 day**          | **5.25**                    | 1024                 |
> | **AIDD (UniCodec)**     | **90M (210M)**      | **1 day**          | 11.53                       | 1024                 |
> | CQT Diff+               | 242M               | 4 days             | 12.54                       | 35                   |
>
>
> **[5] Limited scope of evaluation**
>
> To address concerns regarding evaluation scope, we extended our experiments beyond the music domain and added additional masking scenarios:
>
> -	Evaluation on non-music audio:
> We tested AIDD on the UrbanSound8K dataset (sound effects). For 4-second audio clips with central gaps of 200 ms and 300 ms, we obtained:
>
>     •	200 ms gap: FAD = 0.225, LSD = 0.247
>
>     •	300 ms gap: FAD = 0.381, LSD = 0.276
>
> -	Evaluation under partial masking:
> We also considered partially masked gaps by applying token-level random masking within the missing region. Each token was independently masked with probability p (with p = 1 corresponding to full masking). Results are reported below:
>
>
> | Method                               | FAD ↓ (375 ms) | FAD ↓ (750 ms) | LSD ↓ (375 ms) | LSD ↓ (750 ms) |
> |--------------------------------------|----------------|-----------------|----------------|-----------------|
> | AIDD (Fully masked, p = 1)           | 0.042        | 0.055         | 0.044        | 0.085         |
> | AIDD (Partially masked, p = 0.7)     | 0.035          | 0.052           | 0.042          | 0.085           |
> | AIDD (Partially masked, p = 0.5)     | 0.030          | 0.055           | 0.042          | 0.084           |
>
>
> **[6] Comparison against a continuous latent diffusion baseline**
>
> To further assess the advantage of modeling inpainting using discrete tokens, we conducted an additional experiment comparing AIDD to a latent diffusion baseline operating at the same frame rate:
> | Method                            | FAD ↓ (375 ms) | FAD ↓ (750 ms) | LSD ↓ (375 ms) | LSD ↓ (750 ms) |
> |-----------------------------------|----------------|-----------------|----------------|-----------------|
> | AIDD (Discrete Tokens - 24kHz)            | **0.042**      | **0.055**       | **0.044**      | **0.085**       |
> | Continuous Diffusion (latent - 24kHz)     | 0.207          | 0.281           | 0.068        | **0.084**           |
>
> These results show that the discrete-token formulation provides substantially better reconstruction quality.
>
>
>
> We thank the reviewers for their thoughtful and constructive feedback, which strengthened our work, and we thank the AC for their time and consideration.
>
> Sincerely,
>
> The authors.

---

### Meta-Review · Area_Chair_oDhE · 2026-01-07

**Summary:**

All reviewers were borderline-accept and flagged similar issues about the results being specific to the choice of tokenizer/codec, and lack of subjective listening evidence. There were also some evaluation concerns and the fact that the inpainting scenarios very limited to only central silent gaps. One reviewer explicitly asked for a controlled comparison to continuous-latent diffusion at matched frame rate.

**Reviewer Concerns:**

The rebuttal seems to have addressed most of the core concerns including MOS-style listening test, codec ablations and controls, and comparison to a matched-rate continuous latent diffusion baseline that the authors show to underperform their proposed method. What still remains is generalization beyond “silent-gap” assumptions and testing beyond music. The authors acknowledge current limitations for modeling noisy/non-silent gaps and domain mismatch driven by tokenizer training.

**Reviewer Scores:**

Reviewer jyTi already increased their score after rebuttal, which strongly suggests the rebuttal resolved the major blocking points for at least one reviewer. The other two were already at “would not mind if accepted,” and given that their specific asks for human eval, codec justification, continuous-vs-discrete control, etc., were largely met, I would consider moving the score up.

---

### Decision · Program_Chairs · 2026-01-26

Accept (Poster)